# Selenate Adsorption from Water Using the Hydrous Iron Oxide-Impregnated Hybrid Polymer

**Vesna Marjanovic** [1],*, **Aleksandra Peric-Grujic** [2], **Mirjana Ristic** [2], **Aleksandar Marinkovic** [2], **Radmila Markovic** [1] , **Antonije Onjia** [2] **and Marija Sljivic-Ivanovic** [3]

1  Mining and Metallurgy Institute Bor, Zeleni Bulevar 35, 19210 Bor, Serbia; radmila.markovic@irmbor.co.rs
2  Faculty of Technology and Metallurgy, University of Belgrade, Karnegijeva 4, 11120 Belgrade, Serbia; alexp@tmf.bg.ac.rs (A.P.-G.); risticm@tmf.bg.ac.rs (M.R.); marinko@tmf.bg.ac.rs (A.M.); onjia@anahem.org (A.O.)
3  Vinča Institute of Nuclear Sciences, University of Belgrade, 12-14 Mike Petrovića Street, 11351 Belgrade, Serbia; marijasljivic@vin.bg.ac.rs
*  Correspondence: vesna.marjanovic@irmbor.co.rs; Tel.: +381-69-631736

**Abstract:** Hybrid adsorbent, based on the cross-linked copolymer impregnated with hydrous iron oxide, was applied for the first time for Se(VI) adsorption from water. The influence of the initial solution pH, selenate concentration and contact time to adsorption capacity was investigated. Adsorbent regeneration was explored using a full factorial experimental design in order to optimize the volume, initial pH value and concentration of the applied NaCl solution as a reagent. Equilibrium state was described using the Langmuir model, while kinetics fitted the pseudo-first order. The maximum adsorption capacity was found to be 28.8 mg/g. Desorption efficiency increased up to 70%, and became statistically significant with the reagent concentration and pH increase, while the applied solution volume was found to be insignificant in the investigated range. Based on the results obtained, pH influence to the adsorption capacity, desorption efficiency, Fourier transform infrared (FTIR) and X-ray diffraction (XRD) analysis of loaded adsorbent, it was concluded that the outer- and inner-sphere complexation are mechanisms responsible for Se(VI) separation from water. In addition to the experiments with synthetic solutions, the adsorbent performances in drinking water samples were explored, showing the purification efficiency up to 25%, depending on the initial Se(VI) concentration and water pH. Determined sorption capacity of the cross-linked copolymer impregnated with hydrous iron oxide and its ability for regeneration, candidate this material for further research, as a promising anionic species sorbent.

**Keywords:** macroporous polymer; goethite; factorial design; desorption

## 1. Introduction

Selenium in the environment is naturally present in rocks and soils in a few oxidation states: as selenite, selenate, selenide and elemental Se. It is a trace element in natural deposits of the ore containing the other minerals, such as sulfides of heavy metals [1]. Besides the naturally present Se, its concentration increase in the environment is caused by human activities, especially mining, coal combustion, pesticides production, agriculture, etc. [2]. Nowadays, a large amount of diverse types of wastewater containing harmful chemicals are generated by the industry, and the water crisis is caused by untreated wastewater disposal [3]. Selenium is present in the effluents of the final phases of ore processing, mainly as selenite (at low pH value), or selenate (at high pH value).

Several selenium chemical derivatives such as selenomethionine, selenocysteine, selenate and selenite are the major sources of dietary selenium, out of which selenomethionine is the most widely

consumed [4]. This element is necessary as a micronutrient in the form of selenoproteins, and extremely important for many biological functions, like the formation of thyroid hormones, DNA synthesis, antioxidant defense, fertility and reproduction. Many different classes of naturally and synthetic organoselenium compounds have been explored as the antiproliferative agents, and the field is constantly emerging, with several compounds demonstrating a pronounced cytotoxic activity against cancer cells compared to the non-transformed ones [5]. However, selenium intake in excessive amounts can be extremely toxic for living organisms, depending on the concentration, but also on the chemical form and other dietary components involved. In general, contradictory opinions about its toxicity and necessity caused differences in permitted values of this element in drinking water legislation: due to the World Health Organization [6] and European Commission (EC) [7] the drinking water limit for selenium is 10 $\mu g\,L^{-1}$, while the US Environmental Protection Agency (EPA) standard limit is 50 $\mu g\,L^{-1}$ [8].

Many processes for selenium removal from water, including the chemical reduction, adsorption, bioremediation, phytoremediation, and electrochemical methods, were extensively studied. It is known that sorption plays an important role in transport and control of the target metal contaminants in the ecosystem [9]. Bearing in mind that selenate in soils and sediments preferentially reacts with ferric Fe(III) oxides and hydroxides [10], the synthetic and natural iron-based adsorbents were applied for selenate separation.

Recently, the ER/DETA/FO/FD adsorbent was synthesized in a two step processes: by amination of cross-linked macroporous polymer (ER) with diethylenetriamine (DETA) in the first step and impregnation with hydrous iron oxide (FO), in the second step. The obtained material, after freeze drying (FD) process, has shown a good adsorption potential towards As oxyanions [11]. In this study, the performances of ER/DETA/FO/FD for Se(VI) removal from water were investigated.

The main aim of this work was to investigate the effect of mostly studied parameters: pH, contact time and concentration of selenate to sorption capacity, as well as to examine the possibility of loaded adsorbent regeneration and process optimization by varying desorption reagent pH, volumes and concentrations. Furthermore, the possibility of proposed adsorbent utilization for Se(VI) removal from drinking water was tested as well.

## 2. Materials and Methods

The adsorption experiments were undertaken using the cross-linked macroporous polymer impregnated with hydrous iron oxide, _-FeOOH (ER/DETA/FO/FD). It was previously applied as an efficient adsorbent for arsenate, and its synthesis and characterization were reported previously [11]. The efficiency of the chosen adsorbent for selenium removal has been investigated in the synthetic and real water solution. Synthetic solutions of Se(VI) were prepared by dissolving the appropriate amount of $Na_2SeO_4$ (p.a. purity grade, Sigma Aldrich), in mili-Q water. The real solutions were prepared in the same way as the synthetic solutions, using drinking water instead of mili-Q, in order to simulate drinking water samples with elevated Se(VI) concentration. Drinking water was collected from the water supply network in Belgrade, Serbia.

The adsorption experiments were performed using 4 mg of adsorbent added to 25 mL of synthetic solution in polyethylene flasks, at room temperature (22 °C). The flasks were shaken on the orbital Heldoph shaker (Heidolph North America, Wood Dale, IL, USA) (at a constant speed of 170 rpm. The sorption process was investigated as a function of contact time (15–500 min), initial pH value (2–11) and initial selenium concentration (0.1–5 mg $L^{-1}$), while the temperature and solid/liquid ratio were kept constant.

After defined contact time, in adsorption and desorption experiments, the suspensions were filtered through 0.45 $\mu m$ filter and Se concentration in solution was measured using an inductively coupled plasma mass spectrometry ICP-MS (Agilent 7700 Series, Agilent Technologies, Inc. Tokyo, Japan). In drinking water samples, beside Se, the concentrations of Ca, Mg, Na, K, Pb, Fe, Cu, Zn, Ni, Mg were measured; pH values were monitored using the pH-meter of WTW Ino Lab.

The amount of Se adsorbed (mg g$^{-1}$) at time $t$ ($q_t$) and in the equilibrium ($q_e$) were calculated using Equations (1) and (2), where $c_0$, $c_t$ and $c_e$ are the initial selenium concentration, selenium concentration in solution after appropriate adsorption time ($t$) and in equilibrium ($c_e$), respectively; $V$ is a solution volume and $m$ is an adsorbent mass.

$$q_t = (c_0 - c_t)V/m \tag{1}$$

$$q_e = (c_0 - c_e)V/m \tag{2}$$

The desorbed amount ($Q_{des}$) is calculated as the amount of selenium desorbed from one gram of spent adsorbent (Equation (3)):

$$Q_{des} = c_{des}.V_{des}/M \tag{3}$$

where $c_{des}$ is Se concentration in desorption solution, $V_{des}$ is volume of desorption solution and $M$ is the weight of spent adsorbent. Finally, desorption efficiency (%) is calculated as a ratio of desorbed amount ($Q_{des}$) and initially sorbed amount $q_e$, multiplied by 100.

X-ray diffraction (XRD) analysis and Fourier transform infrared (FTIR) analysis were used to determine the mineralogical and surface composition of Se loaded adsorbent. XRD analysis was undertaken using a small-angle x-ray scattering (SAXS) diffractometer (Rigaku Smartlab, Austin, TX, USA). within 2θ range 10–90 with 0.05 step size. The FTIR analysis has been performed using a Nicolet IS 50 FTIR Spectrometer (Thermo Fisher Scientific, Waltham, MA, USA) operating in the attenuated total reflection (ATR) mode in the region 400–4000 cm$^{-1}$ and resolution of 4 cm$^{-1}$ with 32 scans.

In order to investigate the possibility of adsorbent regeneration, the series of desorption experiments were performed. NaCl, as a non-aggressive reagent, was chosen in the experimental part. The desorption efficiency was investigated as a function of selected process variables: leaching solution concentration, pH value and solution volume (three levels), using a full factorial experimental design (Table 1). A high level of volume was 25 mL as in the adsorption experiment while the lower volumes were tested in order to investigate the desorption process that generates a lower amount of waste liquid.

**Table 1.** Process variables and their levels.

| Variable Code | Variable | Levels | | |
| --- | --- | --- | --- | --- |
| | | Low | Medium | High |
| A | Initial pH | 7 | - | 11 |
| B | Concentration (mol L$^{-1}$) | 0 | - | 0.5 |
| C | Volume (mL) | 5 | 15 | 25 |

The experimental design matrix was generated using the Minitab software (Released 13.0, State College, PA, USA). The values of process variables were specified in each experiment. The aqueous solutions, used in these experiments, were obtained using mili-Q water and NaCl and NaOH (p.a. purity). The other experimental conditions were constant: temperature (22 °C), agitation rate (170 rpm) and contact time (300 min). The experiments were undertaken in triplicate and the average values of response function were considered in the statistical analysis. The experiments were performed in a random order to assure that the uncontrolled factors do not affect the results and to evaluate the experimental errors properly. The results interpretation and analysis were also undertaken using the Minitab software.

## 3. Results

### 3.1. Effect of pH

Investigation of the effect of initial pH value (pH$_i$) onto Se sorption efficiency has shown that it increases the efficiency (Figure 1) at pH between 2 and 4. Further increase in pH resulted in a significant decrease of sorption efficiency.

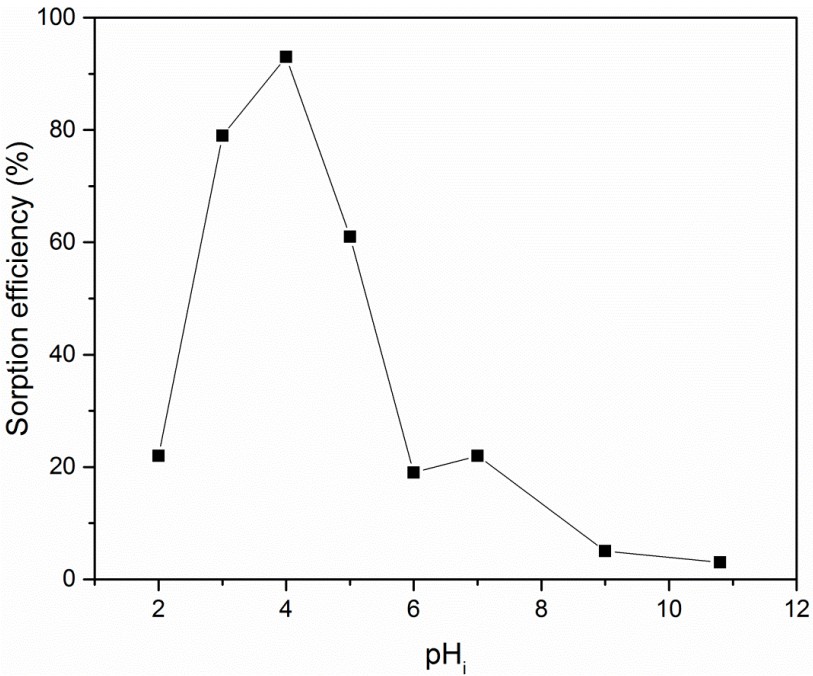

**Figure 1.** The effect of initial pH value (pH$_i$) onto sorption efficiency (%); experimental conditions: $c_0 = 1$ mg L$^{-1}$, temperature 22 °C, stirring rate 170 rpm, contact time 300 min.

### 3.2. X-ray Diffraction (XRD) Analysis

The results of XRD analysis of the ER/DETA/FO/FD before adsorption and Se loaded ER/DETA/FO/FD are shown in Figure 2. Peaks, characteristic for goethite, are observed at the 2$\theta$ value of 21.2, 33.2, 36.6 and 53.2, referring to the ICDD PDF2 No. 81-0464. Similar peaks were observed after adsorption, indicating that the crystallinity of material did not significantly change due to the adsorption of selenate.

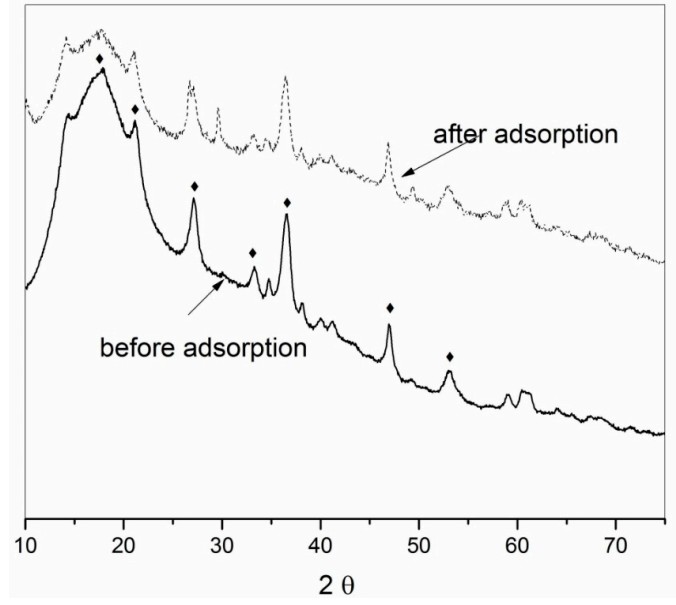

**Figure 2.** X-ray diffraction (XRD) records of the ER/DETA/FO/FD adsorbent before and after adsorption (experimental conditions for preparation of Se loaded sample: Se initial concentration 5 mg L$^{-1}$, ambient temperature, solid/liquid ratio 0.16 g L$^{-1}$, contact time 300 min). Symbol (♦) denotes characteristic peaks of goethite.

### 3.3. Process Kinetics

The dependence of sorption capacity on a contact time is shown in Figure 3. The removal was faster at the beginning of the process what can be attributed to the higher number of free active sites, as well as to the more intensive driving force for the mass transfer. The sorption capacity obtained in equilibrium was around 5.8 mg g$^{-1}$ which is more than 90% of the total Se(VI) amount. The equilibration was attained after 300 min.

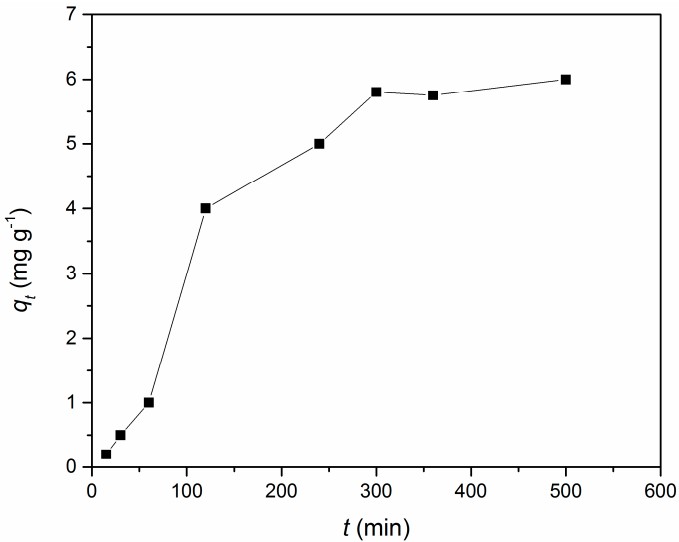

**Figure 3.** The effect of contact time onto Se(VI) adsorption using the ER/DETA/FO/FD; experimental conditions: $c_o$ = 1 mg L$^{-1}$, temperature 22 °C, stirring speed 170 rpm, pH = 4, sorbent dose 0.16 g L$^{-1}$.

### 3.4. Effect of the Initial Se(VI) Concentration—Adsorption Isotherms

The effect of the initial adsorbate concentration on the adsorbed Se(VI) amount was studied; the initial Se concentrations varied in the range of 0.1 to 5 mg L$^{-1}$. With the increase of initial Se(VI) concentration in solution, the sorbed amount in the solid phase increased, as well as the equilibrium concentration (Figure 4). According to the Gils classification [12], the isotherm obtained is of the L-type.

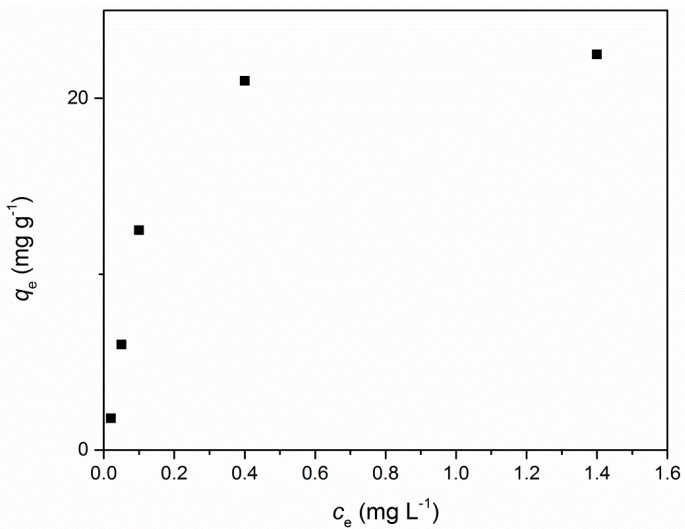

**Figure 4.** Adsorption isotherm for selenate by the ER/DETA/FO/FD adsorbent at pH = 4; contact time 300 min, initial concentration range from 0.1 to 5 mg/L, sorbent dose 0.16 g L$^{-1}$.

### 3.5. Fourier Transform Infrared (FTIR) Spectra of Adsorbent

In order to analyze the surface of material after sorption, the FTIR spectra of Se unloaded and loaded adsorbent were recorded (Figure 5). The broad band at ~3370 cm$^{-1}$ belongs to the hydroxyl stretching region. It is more intense on loaded adsorbent, due to the adsorption of hydroxyl ions. In the region 1100–2000 cm$^{-1}$, the position of bands after adsorption has not been significantly changed; nevertheless, some bands became more intense. Compared to the spectrum a), two new bands at 950 and 852 cm$^{-1}$ appeared, and they can be assigned to the bands of adsorbed SeO$_4$ species. The sharp peak at 852 cm$^{-1}$ is similar to the spectra of selenate in solution [13], indicating an outer-sphere complex present at the adsorbent–solution interface; an oxyanion retains its hydration shell and does not form a direct chemical bond with the surface, but a complex involving electrostatic forces.

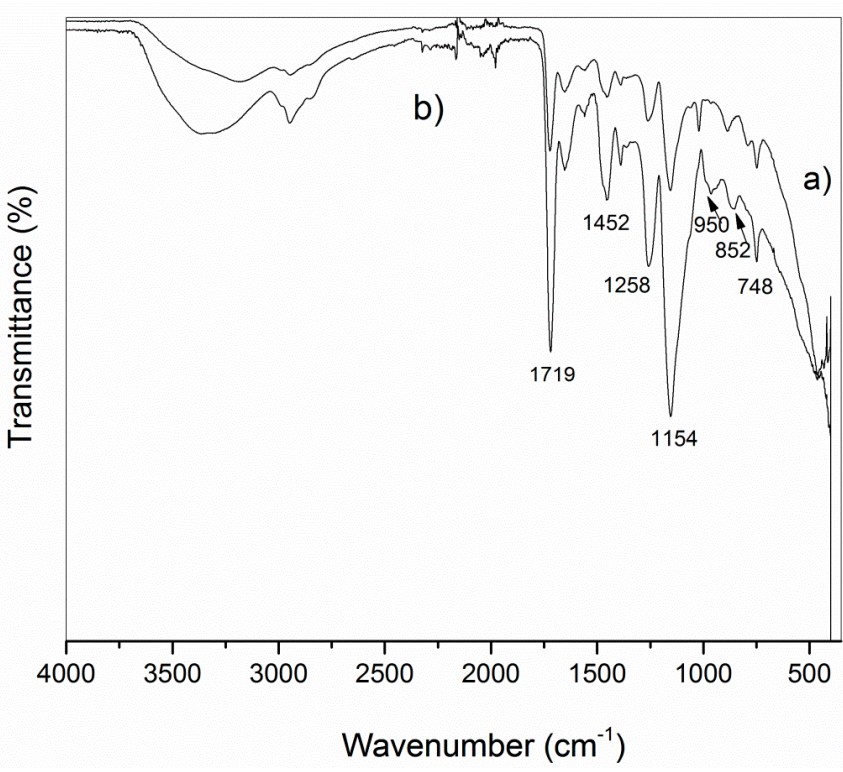

**Figure 5.** Fourier transform infrared (FTIR) spectrum of adsorbent (**a**) before and (**b**) after Se adsorption.

### 3.6. Regeneration Studies

Since a high sorption capacity of the investigated sorbent was achieved, the regeneration studies were performed in order to determine the possibility of its reuse The full factorial experimental design was used in this study, since this method has proven to be suitable for the adsorption process investigations [14,15]. The effect of desorption solution pH, concentration and volume onto desorption efficiency was studied. The experimental design matrix was generated on the basis of two levels of factor (pH and concentration) and three levels of solution volume and comprised 12 runs (Table 2).

Desorption efficiency (Figure 6) was insignificant when the mili-Q water, pH = 7 was used (runs 7, 8, 9). The process was more efficient (50–70%) when solutions with a higher pH or NaCl concentration were used.

**Table 2.** Experimental design matrix.

| Run | *A* | *B* | *C* | Initial pH | *C* (NaCl) mol L$^{-1}$ | *V* (mL) | Final pH |
|---|---|---|---|---|---|---|---|
| 1 | 2 | 2 | 3 | 11 | 0.5 | 25 | 10.9 |
| 2 | 1 | 2 | 1 | 7 | 0.5 | 5 | 7.0 |
| 3 | 2 | 2 | 1 | 11 | 0.5 | 5 | 10.8 |
| 4 | 2 | 1 | 3 | 11 | 0 | 25 | 11.1 |
| 5 | 2 | 1 | 1 | 11 | 0 | 5 | 10.5 |
| 6 | 2 | 2 | 2 | 11 | 0.5 | 15 | 10.7 |
| 7 | 1 | 1 | 1 | 7 | 0 | 5 | 6.2 |
| 8 | 1 | 1 | 3 | 7 | 0 | 25 | 6.6 |
| 9 | 1 | 1 | 2 | 7 | 0 | 15 | 6.6 |
| 10 | 1 | 2 | 2 | 7 | 0.5 | 15 | 6.8 |
| 11 | 2 | 1 | 2 | 11 | 0 | 15 | 10.9 |
| 12 | 1 | 2 | 3 | 7 | 0.5 | 25 | 6.9 |

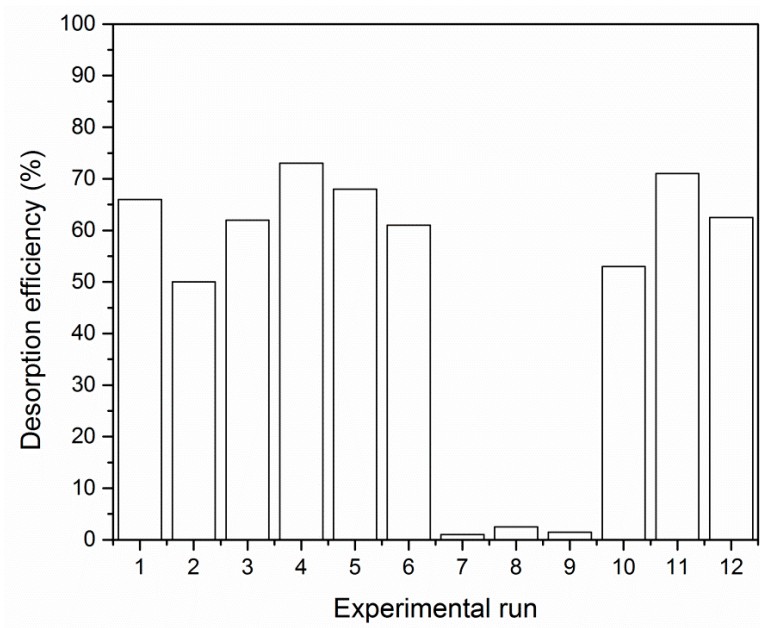

**Figure 6.** Desorption efficiency obtained for each experimental run (loaded adsorbent from adsorption studies after equilibration with Se, $c_o$ 5 mg L$^{-1}$). The experimental conditions of each run are given in Table 2.

### 3.7. Separation of Selenate from Spiked Drinking Water Samples

In this study, series of spiked drinking water samples were prepared with different Se concentrations (1 mg L$^{-1}$, 2 mg L$^{-1}$ and 5 mg L$^{-1}$) and initially adjusted pH values at 4 and 7. Concentrations of selected cations are presented in Table 3.

**Table 3.** Concentrations of selected metal cations in drinking water samples (mg L$^{-1}$).

| Ca | Mg | K | Fe | Pb | Cu | Zn | Ni | Mn |
|---|---|---|---|---|---|---|---|---|
| 49.1 ± 4.5 | 8.8 ± 0.8 | 4.8 ± 0.5 | <0.01 | <0.02 | 0.15 ± 0.03 | <0.05 | <0.01 | <0.01 |

The concentration of analyzed cations did not change during adsorption due to a repulsion of these ions by a positively charged sorbent surface. The adsorption experiments with drinking water at pH 4 have shown that the removal efficiency of applied adsorbent was significantly lower compared to the efficiency obtained using Se(VI) in synthetic (distilled) water (Table 4).

**Table 4.** The removal efficiency of selenate by the ER/DETA/FO/FD adsorbent from drinking water.

| Selenate Solution in | pH | Se initial Concentration (mg L$^{-1}$) | Removal Efficiency (%) |
|---|---|---|---|
| Synthetic water | 4 | 1 | 91 |
| | | 2 | 77.5 |
| | | 5 | 71 |
| Drinking water | 4 | 1 | 25 |
| | | 2 | 17.5 |
| | | 5 | 13 |
| Drinking water | 7 | 1 | 14 |
| | | 2 | 8 |
| | | 5 | 6.5 |

## 4. Discussion

The pH value of solution is an important factor in adsorption processes since it affects the metal species distribution, as well as the protonation of the adsorbent surface groups. Selenium(VI) distribution as a function of pH, obtained using the MINTEQ software, showed a coexistence of $HSeO_4^-$ and $SeO_4^{2-}$ at pH < 4, while at pH ≥ 4 $SeO_4^{2-}$ is dominantly present in solution. Even the surface of adsorbent, studied in this research, is positively charged until $pH_{pzc}$ 8.8 [11], and a highest sorption capacity was obtained at pH 4 (Figure 1). In order to explain this phenomenon, the reactions taking place at the adsorbent surface (cross-linked copolymer modified with hydrous ferric oxide, mainly in the form of goethite (Figure 2), are presented with the appropriate constants ($K_{a1}$ and $K_{a2}$) [16]. It is obvious that both of them are pH dependent.

$=$ Fe-OH is neutral, while $=$ FeOH$_2^+$ and $=$ Fe-O$^-$ are in their protonated and deprotonated forms, respectively.

$$= \text{FeOH}_2{}^+ \rightarrow \text{H}^+ + \; = \text{Fe} - \text{OH} \cdots K_{a1} = \frac{\left[\text{H}^+\right]\left[= \text{FeOH}\right]}{\left[= \text{FeOH}_2^+\right]} = 10^{-6.5}$$

$$= \text{FeOH}_2{}^+ \rightarrow \text{H}^+ + \; = \text{Fe} - \text{O} \cdots K_{a2} = \frac{\left[\text{H}^+\right]\left[= \text{FeO}^-\right]}{\left[= \text{FeOH}\right]} = 10^{-9}$$

At pH = 4, FeOH$_2^+$ is dominant at solid surface [16] and $SeO_4^{2-}$ in the liquid phase; with the pH increase, in the pH range 5–8, FeOH$_2^+$ concentration decreases, as well as the adsorbent sorption capacity for selenate ions. In addition, the adsorption of Se(VI) on to a positively charged adsorbent is diminished in a highly acidic media, at pH < 2 (Figure 1) due to the dominantly occurring $HSeO_4^-$ ion. Based on the results obtained, it can be concluded that $SeO_4^{2-}$ ions are bonded to FeOH$_2^+$ present on the surface.

The relation between $q_t$ and time ($t$) was analyzed using the well-known kinetic models in the linear forms:

Pseudo-first order [17]:

$$ln\frac{(q_e - q_t)}{q_e} = -k_1 \cdot \text{t} \tag{4}$$

Pseudo-second order [18]:

$$\frac{t}{q_t} = \frac{1}{k_2 \cdot q_e^2} + \frac{t}{q_e} \tag{5}$$

Intraparticle diffusion model [19]:

$$q_t = k_d \cdot t^{0.5} + c \tag{6}$$

where $k_1$, $k_2$ and $k_d$ denote constants of the pseudo-first, pseudo-second and intraparticle diffusion model, respectively. The parameters of the model were calculated from the slope-intercept form. Furthermore, the determination coefficient ($R^2$), *F*-value and *p*-value were calculated in order to evaluate the accuracy of the applied models (Table 5).

**Table 5.** Kinetic model parameters.

| Model | Parameters | | | | | |
|---|---|---|---|---|---|---|
| Pseudo-second order model | $q_e$ 13.99 mg g$^{-1}$ | $h$ $3.54 \times 10^{-2}$ mg (g min)$^{-1}$ | $k_2$ $2.53 \times 10^{-3}$ g (mg min)$^{-1}$ | $R^2$ 0.651 | $F$ 7.52 | $p$ 0.052 |
| Pseudo-first order model | $q_e$ 7.24 mg g$^{-1}$ | $k_1$ $9.16 \times 10^{-3}$ min$^{-1}$ | | 0.986 | 363 | $7.3 \times 10^{-6}$ |
| Intraparticle diffusion model | $k_d$ 0.374 mg g$^{-1}$ min$^{-1/2}$ | | | 0.952 | 100 | $1.7 \times 10^{-4}$ |

Higher $R^2$ and *F*-value and lower *p*-value ($p < 0.05$) could point out the models which are suitable for data description. Also, the experimentally determined $q_e$ value and that obtained from the plot of $t/q_t$ vs. $t$ differ significantly, indicating that the process mechanism does not follow the pseudo-second order model, but the pseudo-first. The plot of $q_t$ versus $t^{0.5}$ (Equation (6)) is linear, but does not pass through the origin, indicating that an intra-particle diffusion is not the mechanism of sorption [19].

The maximum sorption capacity, defined from the plateau part, was 22.5 mg g$^{-1}$. Sorption isotherms were fitted using the most utilized models, Langmuir and Freundlich. One of the mostly used linear forms of the Langmuir model [20] is:

$$\frac{c_e}{q_e} = \frac{c_e}{q_m} + \frac{1}{q_m K_L} \tag{7}$$

while, the linearization of the Freundlich [21] model gives:

$$ln q_e = ln K + 1/n \cdot ln c_e \tag{8}$$

where $q_e$ (mg g$^{-1}$) and $c_e$ (mg L$^{-1}$) denote the equilibrium concentration of ions in the solid and liquid phase, respectively, $q_m$ (mg g$^{-1}$) is the maximum sorption capacity, $K_L$ (L g$^{-1}$) is the Langmuir constant related to the energy of adsorption, $n$ and $K_F$ are the Freundlich isotherm parameters.

Linear fitting of functions $c_e/q_e$ vs. $c_e$, and $ln q_e$ vs. $ln c_e$ gave the parameters of the Langmuir and Freundlich models, respectively (Table 6).

**Table 6.** Parameters of the Langmuir and Freundlich isotherm model, obtained for selenate adsorption by the ER/DETA/FO/FD.

| Model | Model Parameter | Value |
|---|---|---|
| Langmuir | $K_L$ | 2.981 L mg$^{-1}$ |
| | $q_m$ | 28.8 mg g$^{-1}$ |
| | $F$ | 6.69 |
| | $p$ | 0.081 |
| | $R^2$ | 0.78 |
| Freundlich | $n$ | 1.246 |
| | $K_F$ | 30.377 mg$^{1-n}$L$^{3n}$g$^{-1}$ |
| | $F$ | 5.62 |
| | $p$ | 0.14 |
| | $R^2$ | 0.736 |

Additionally, the higher *F*- and $R^2$ values, as well as a lower *p*-value have shown that the Langmuir model is more suitable for the experimental data description than the Freundlich model, indicating that a homogeneous surface of adsorbent is covered with a single layer of adsorbed molecules [22].

The effect of process variables (*A*—initial pH, *B*—NaCl concentration, *C*—desorption solution volume) onto system responses (percentage of the desorbed amounts and final pH values) were evaluated using the statistical software. Also, the interactions between variables were considered, since they might have a significant effect on system response. The coefficients in the equation were calculated using the second order regression model (Equation (9)), giving the information about the effect of process variables.

$$Y = \beta_1\, A + \beta_2\, B + \beta_3\, C + \beta_{12}\, AB + \beta_{13}\, AC + \beta_{23}\, BC + \varepsilon, \tag{9}$$

where: *Y*—system response; *A*, *B*, *C*—independent variables, meaning the initial pH, salt concentration and volume of desorption solution; *AB*, *AC*, *BC*—interactions terms; $\beta_1$, $\beta_2$, $\beta_3$, $\beta_{12}$, $\beta_{13}$, $\beta_{23}$ —regression coefficients; $\varepsilon$—residual.

In addition, the analysis of variance (ANOVA) was used in order to define the statistically significant factors and/or their interactions (Table 7). The results have shown that a desorption efficiency was significantly influenced by the factors *A* and *B*, and their interaction (*AB*) at $p < 0.05$. On the other hand, the volume of the leaching reagent did not play any important role in the investigated desorption process.

**Table 7.** Analysis of variance (ANOVA) regression analysis for Se(VI) desorption data.

| Variable | System Response | | | |
|---|---|---|---|---|
| | Desorption (%) | | Final pH | |
| | *F* | *p* | *F* | *p* |
| *A* | 1574 | 0.001 | 20195 | <0.001 |
| *B* | 634.4 | 0.002 | 48.48 | 0.02 |
| *C* | 16.89 | 0.056 | 21.20 | 0.045 |
| *AB* | 944.0 | <0.001 | 63.84 | 0.015 |
| *AC* | 0.19 | 0.839 | 2.84 | 0.260 |
| *BC* | 6.68 | 0.13 | 43.00 | 0.030 |
| - | *s* = 1.691 $R^2$ = 99.94% $R^2$ (adj) = 99.66% | | *s* = 0.0505 $R^2$ = 99.99% $R^2$ (adj) = 99.95% | |

Considering the final pH values as a system response, all of the investigated variables and their interactions were statistically significant except the *AC* interaction. The obtained $R^2$ values were >99.9%, indicating that the experimental data for both system responses could be explained with a high accuracy.

Visualization of the results obtained was undertaken using the *Main effect plots* and *3D surface plots* (Figure 7). The increase of each process variable provoked the increase of system response. The effect of initial pH of leaching solution had the highest effect on desorption efficiency and final pH.

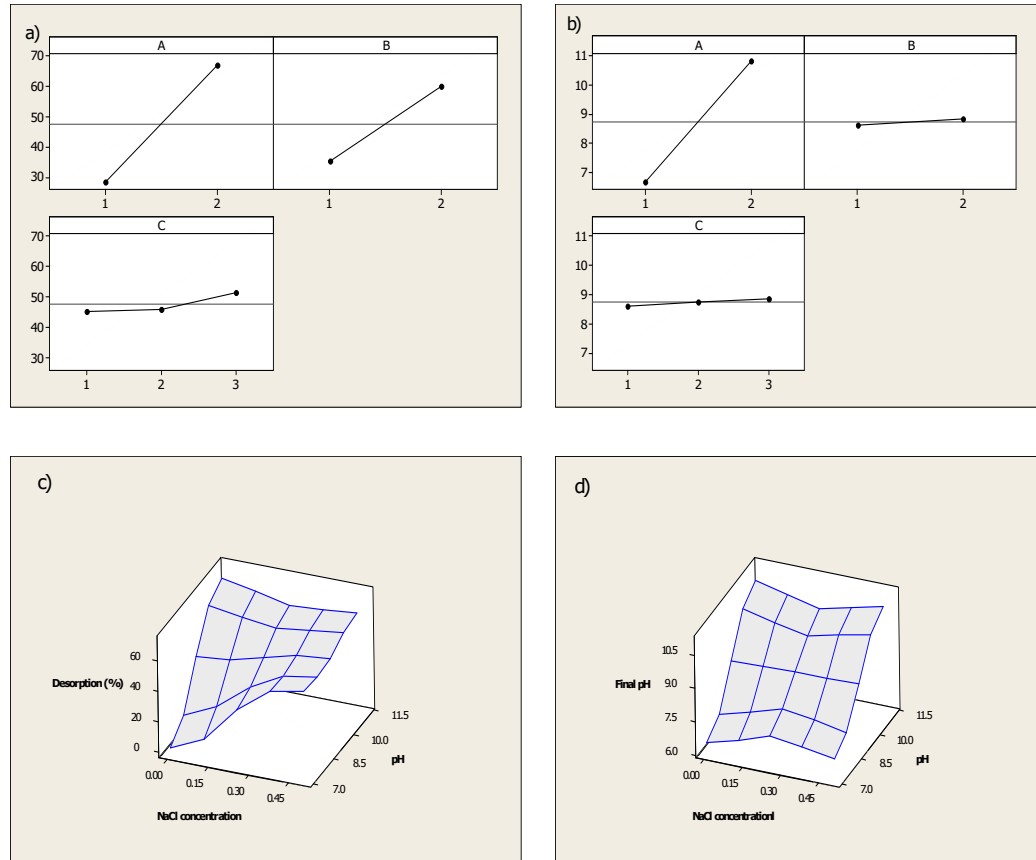

**Figure 7.** The main effect plots for (**a**) desorption efficiency and (**b**) final pH. Three-dimensional surface plots for (**c**) desorption efficiency and (**d**) final pH.

The removal efficiency of applied adsorbent was lower in spiked drinking water solutions, compared to the adsorption from distilled water (pH 4), Table 4. This is in line with the reported results where a significant adsorption suppression was observed when phosphate and sulphate anions coexisted [15,23]. In part, a non-specific sorption through electrostatic attraction between the negatively charged anions and positive sites on the sorbent obviously takes place. Efficiencies at pH = 7 were about 50% lower in comparison with those obtained at pH = 4.

On the basis of the summarized experimental results, the mechanism of selenate adsorption by the cross-linked macroporous polymer impregnated with hydrous iron oxide (ER/DETA/FO/FD) is proposed. The highest removal efficiency and maximum adsorption capacity were obtained at pH = 4, due to the formation of a complex between $FeOH_2^+$ at the solid surface and $SeO_4^{2-}$, dominantly present in the liquid phase at this pH. The maximum observed desorption efficiency was 72.5%, at pH = 11, indicating that about 30% of selenate is irreversibly adsorbed, probably due to the formation of the inner-sphere complex (Equation (10)), known as incompletely reversible. On the other hand, the outer-sphere complexation (Equations (11) and (12)), largely electrostatic, is a reversible process responsible for desorption. In the outer-sphere complex, the ion retains its hydration sphere and attaches to the surface via electrostatic forces, whereas the inner-sphere complex is partially dehydrated and directly bound to the surface [24].

$$>FeOH + H^+ + SeO_4^{2-} \rightarrow >FeSeO_4^- + H_2O \qquad (10)$$

$$>FeOH + H^+ + SeO_4^{2-} \rightarrow >FeOH_2^+ - SeO_4^{2-} \qquad (11)$$

$$2 > FeOH + 2H^+ + SeO_4^{2-} \rightarrow (>FeOH_2^+)_2 - SeO_4^{2-} \qquad (12)$$

A possible schematic presentation of the adsorbed selenate species on the FEG–SEM (field-emission gun—scanning electron microscopy) image of ER/DETA/FO/FD is given in Figure 8.

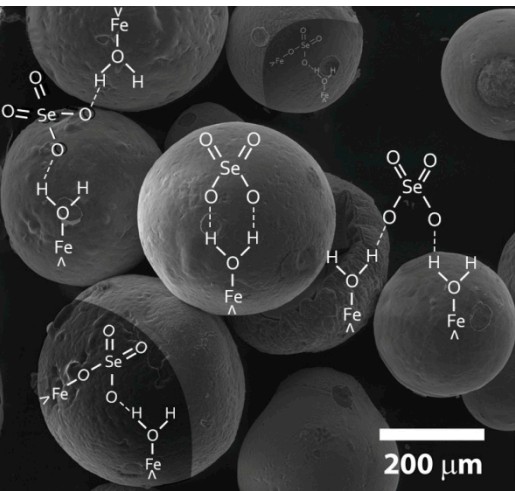

**Figure 8.** Structures of selenate surface complexes on macroporous polymer impregnated with hydrous iron oxide (ER DETA/FO/FD).

## 5. Comparative Evaluation of Different Adsorbents for Se(VI) Removal

The adsorbent investigated in this study is compared with the iron oxide-modified adsorbents applied for the removal of selenate from water (Table 8). It is obvious that the adsorption capacity of material investigated has a higher value than those obtained using the other adsorbents.

**Table 8.** Experimental conditions and uptake capacities of various iron containing sorbents for the selenate removal from water.

| Adsorbent | Experimental Conditions | | | | | | Maximum Sorption Capacity (mg g$^{-1}$) | Reference |
|---|---|---|---|---|---|---|---|---|
| | pH | Se(VI) Concentration Range | Adsorbent Dosage | Volume (ml) | Contact Time (min) | Temperature | | |
| Natural hematite from Cerro del Hierro (Spain) | 4 | $3 \times 10^{-6}$ and $5 \times 10^{-4}$ mol dm$^{-3}$ | 100 mg | 20 | - | room | 0.18 | [14] |
| Natural goethite from Cerro del Hierro (Spain) | 4 | $3 \times 10^{-6}$ and $5 \times 10^{-4}$ mol dm$^{-3}$ | 100 mg | 20 | - | room | 0.24 | [14] |
| Synthetic Jacobsite (MnFe$_2$O$_4$) NM | 4 | 0.25–10 mg L$^{-1}$ | 10 mg | 4 | 15 | room | 0.76 | [25] |
| Fe$_3$O$_4$ nanomaterials produced by non microwave-assisted synthetic techniques | 4 | 0.25–10 mg L$^{-1}$ | 10 mg | 4 | 15 | room | 1.43 | [15] |
| Fe$_3$O$_4$ nanomaterials produced by non microwave-assisted synthetic techniques | 4 | 0.25–10 mg L$^{-1}$ | 10 mg | 4 | 15 | room | 2.37 | [15] |
| Iron (Fe$^{3+}$) oxide/hydroxide nanoparticles sol (NanoFe) | 4 | 12 ppm | 15–635 mg L$^{-1}$ | - | 1 | room | 15.1 | [26] |
| Low-Cost Goethite Nanorods | 7.2 | ~0.500 mg L$^{-1}$ | 0.05–1 g L$^{-1}$ | 100 | 360 | room | 4.75 | [27] |
| Iron oxide impregnated hybrid polymer | 4 | 0.1–5 mg L$^{-1}$ | 0.16 g L$^{-1}$ | 25 | 300 | room | 28.8 | This study |

## 6. Conclusions

In this study, the adsorption properties of cross-linked macroporous polymer impregnated with hydrous iron oxide for the removal of Se(VI) ions were tested. Based on the results concerning the adsorption processes, the following conclusions were drawn:

- the investigated process was pH dependent, with the best performances at pH 4;
- a pseudo-first model was the most appropriate for the kinetic data description;
- experimentally determined maximum adsorption capacity of the investigated adsorbent towards Se(VI) was found to be 22.5 mg/g, while the value calculated using the Langmuir model was 28.8 mg/g, depicting its prominent adsorption potential.

Furthermore, the desorption process of adsorbed Se(VI) ions was investigated using the full factorial design as a function of leaching solution pH, NaCl concentration and applied volume. It was observed that the most important factor for desorption efficiency was the interaction between solution pH and concentration, followed by these factors solely. The effect of used leaching solution volume was insignificant in the investigated range. The increase of solution concentration and/or pH, increased the desorption efficiency up to 70% showing that the adsorbent is partly regenerative. The mechanism of selenate sorption by sorbent examined in this research probably includes the outer- and inner-sphere surface complex formation and the process rate determines the inner-sphere formation as a slower one.

**Author Contributions:** Individual contributions of authors are as following: conceptualization, M.S.-I. and A.M.; methodology, M.R., A.P.-G.; software, A.O.; validation, V.M. and R.M.; formal analysis, V.M., R.M. and A.O.; investigation, V.M. and M.R.; resources, A.M., V.M. and R.M.; data curation, V.M.; writing—original draft preparation, V.M. and M.S.-I; writing—review and editing, M.R. and A.P.-G.; visualization, M.R. and A.P.-G.; supervision, M.S.-I.; project administration, M.R. All authors have read and agreed to the published version of the manuscript.

**Funding:** This research received no external funding.

**Acknowledgments:** Financial support for this study was partly provided by the Ministry of Education, Science and Technological Development of the Republic of Serbia (Contract No. 451-03-68/2020-14/200135).

**Conflicts of Interest:** The authors declare no conflict of interest.

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
