# Peer review of "Selenate Adsorption from Water Using the Hydrous Iron Oxide-Impregnated Hybrid Polymer"

_metals, doi:10.3390/met10121630_

Round 1

Reviewer 1 Report

The submitted manuscript (metals-10014468) untitled "Selenate adsorption from water using 2 iron oxide impregnated hybrid polymer" has been reviewed. In this work, The authors reported that a cross-linked macroporous polymer impregnated with hydrous iron oxide was applied for Se (VI) adsorption from water. The quality of the paper should be improved in English and in depth. Also, synthesis method is not novel based on the reported refs. And application part is not supported by strong characterization, and systematic study of selenate adsorption is not carried out. I do not recommend that this manuscript could be published in the "Metals".

  1. The quality of abstract should be significantly improved.
  2. Some characterization should be performed to confirm the structure of the adsorbent.
  3. The result of FTIR and XPS after the adsorption of Se (VI) must be given to make sure the adsorption mechanism.
  4. XRD data, adsorption kinetics and adsorption thermodynamics data need to be reprocessed, kinetic and thermodynamic adsorption experiments should be provided at different temperatures.
  5. A comparison of some recently reported refs on Se (VI) adsorption should be provided, confirming the application prospect of the adsorbent.

Author Response

Dear Reviewer 1,

The authors thank the reviewers for their careful reading, thoughtful and helpful comments on the manuscript. We have carefully taken their comments into consideration in preparing our revision, which has resulted in a paper that is clearer and more compelling. The following summarizes how we responded to the reviewer comments.

Kind regards,

Authors.

Reviewer 2 Report

Review on SELENATE ADSORPTION FROM WATER USING IRON OXIDE IMPREGNATED HYBRID POLYMER

Authors - Vesna Marjanovic * , Aleksandra Peric-Grujic , Mirjana Ristic , Aleksandar Marinkovic , Radmila Markovic , Antonije Onjia , Marija Sljivic-Ivanovic

The paper is interesting but should be improved.  In this form I recommend major revisions.

Please consider these queries

L 37 – “provoked” should be replaced with “caused”

L 42-44 – Please insert bibliography regarding Se derivatives functions

L 66 – hydrous iron oxide – write the formula

XRD analysis – the patterns  are overlapped and should be presented apart, to be easy to see the profile of the spectra and the peaks; also the peaks should be attributed. Why the powder has a low crystallinity degree?

L 176 – after 852 add cm-1

FTIR – present the spectra before and after the adsorption

L 238 – please verify the reaction

The information from Fig 7 should be transposed into a table

In the discussion section should be presented a comparative study of the obtained results and the previous study from the literature.

The manuscript should be revised by a native English speaker.

Author Response

Dear  Reviewer 2,

The authors thank the reviewers for their careful reading, thoughtful and helpful comments on the manuscript. We have carefully taken their comments into consideration in preparing our revision, which has resulted in a paper that is clearer and more compelling. The following summarizes how we responded to the reviewer comments.

Kind regards,

Authors

Round 2

Reviewer 1 Report

The current revision meets the requirements of the journal

Reviewer 2 Report

In in this form the manuscript can be accepted for publication.